# Bioactive Compounds in Anti-Diabetic Plants: From Herbal Medicine to Modern Drug Discovery

**DOI:** 10.3390/biology9090252

**Published:** 2020-08-28

**Authors:** Ngan Tran, Bao Pham, Ly Le

**Affiliations:** 1School of Biotechnology, International University—Vietnam National University, Ho Chi Minh City 721400, Vietnam; kimnganchemistry@gmail.com; 2Information Science Faculty, Saigon University, Ho Chi Minh City 711000, Vietnam; ptbao2005@gmail.com

**Keywords:** diabetes mellitus, medicinal plants, herb, bioactive compound

## Abstract

Natural products, including organisms (plants, animals, or microorganisms) have been shown to possess health benefits for animals and humans. According to the estimation of the World Health Organization, in developing countries, 80% of the population has still depended on traditional medicines or folk medicines which are mostly prepared from the plant for prevention or treatment diseases. Traditional medicine from plant extracts has proved to be more affordable, clinically effective and relatively less adverse effects than modern drugs. Literature shows that the attention on the application of phytochemical constituents of medicinal plants in the pharmaceutical industry has increased significantly. Plant-derived secondary metabolites are small molecules or macromolecules biosynthesized in plants including steroids, alkaloids, phenolic, lignans, carbohydrates and glycosides, etc. that possess a diversity of biological properties beneficial to humans, such as their antiallergic, anticancer, antimicrobial, anti-inflammatory, antidiabetic and antioxidant activities Diabetes mellitus is a chronic disease result of metabolic disorders in pancreas β-cells that have hyperglycemia. Hyperglycemia can be caused by a deficiency of insulin production by pancreatic (Type 1 diabetes mellitus) or insufficiency of insulin production in the face of insulin resistance (Type 2 diabetes mellitus). The current medications of diabetes mellitus focus on controlling and lowering blood glucose levels in the vessel to a normal level. However, most modern drugs have many side effects causing some serious medical problems during a period of treating. Therefore, traditional medicines have been used for a long time and play an important role as alternative medicines. Moreover, during the past few years, some of the new bioactive drugs isolated from plants showed antidiabetic activity with more efficacy than oral hypoglycemic agents used in clinical therapy. Traditional medicine performed a good clinical practice and is showing a bright future in the therapy of diabetes mellitus. World Health Organization has pointed out this prevention of diabetes and its complications is not only a major challenge for the future, but essential if health for all is to be attained. Therefore, this paper briefly reviews active compounds, and pharmacological effects of some popular plants which have been widely used in diabetic treatment. Morphological data from V-herb database of each species was also included for plant identification.

## 1. Introduction

Diabetes mellitus (DM) is a metabolic disorder of multiple causes characterized by chronic hyperglycemia with disturbances of carbohydrate, fat and protein metabolism resulting from defects in insulin secretion, insulin action, or both. The effects of diabetes mellitus include long–term damage, dysfunction, and failure of various organs [1]. Diabetes mellitus is divided into three main types [1,2,3,4,5]. Type 1 diabetes (insulin-dependent diabetes mellitus) is an autoimmune disorder developing when insulin-producing cells of the pancreas in the body have been destroyed and the pancreas produces little or no insulin. A person who has type 1 DM must take insulin daily to live. It develops most often in children and young adults. Type 2 diabetes has also been known as another term “insulin-independent diabetes mellitus” which accounted for more than 90% of diagnosed cases of DM in adults. It is a diagnosis in which the pancreas produces enough insulin but the body cannot use the insulin effectively, a condition called insulin resistance. Gestational diabetes mellitus (GDM) is a degree of glucose intolerance with onset or first recognition in the second or third trimester of the period of pregnancy. GDM is caused by the hormone of pregnancy or a shortage of insulin. GDM is one of the most popular disorders of metabolism during pregnancy.

Hyperglycemia causes damage to eyes, kidneys, nerves, heart and blood vessels [6]. According to the ninth edition 2019 of the International Diabetes Federation (IDF) Diabetes Atlas released by the IDF, as of 2019, the total adult population in the age group of 20–79 years stands at 463 million who live with diabetes, which is set to increase to 578 million by 2030 (Figure 1) [7]. There is one patient who dies of diabetes mellitus every 6 s, this rate is higher than death rates from human immunodeficiency virus (HIV) (1.5 million), tuberculosis (1.5 million) and malaria (0.6 million), combined [8,9,10].

Following the Statistics of International Diabetes Federation in 2019 [7], the total adult population in the age group of 20–79 years stands at 463 million live with diabetes, which may increase to 578 million by 2030. Among them, in 2019, 373.9 million adults aged 20–79 years worldwide, 7.5% of the adult population, are estimated to have impaired glucose tolerance. Most adults with impaired glucose tolerance are under the age of 50 years (180.0 million–48.1%). The estimated prevalence of diabetes in men aged 20–79 years is slightly higher than in women (9.6% vs 9.0%). The prevalence of diabetes is expected to increase in both men and women from 2019 to 2030 and 2045 (Figure 2). All around the world, there are top 10 countries that have numbers of people with diabetes, including China, the USA, Indonesia, India, Brazil, Mexico, Japan, Pakistan, Thailand, and Nigeria. Although the risks of GDM with pregnant women have been recognized clearly, there is uncertainty that the treatment reduces and controls blood glucose level of women during pregnancy could decrease those risks or not. Moreover, GDM extends to increase the risk for the development of type 2 DM after giving birth. Nowadays, diabetes mellitus has risen along with rapid cultural and social changes, such as aging, population, less physical activities, dietary, and so on. The cost associate with diabetes includes increasing health services, an economic burden.

## 2. Management of Diabetes Mellitus

Understanding the pathogenesis of diabetes mellitus is extremely important in treatment. The American Diabetes Association has recommended that the purpose of glycemic control includes a pre-prandial blood glucose level of 80 to 120 mg/dL (4.4 to 6.7 mmol/L), a bedtime blood glucose level of 100 to 140 mg/dL (5.6 to 7.8 mmol/L), and an HbA1c level of less than 7% [2,11]. Healthy eating, physical activity, and weight control are the center of any therapeutic program for patients in DM [12]. These lifestyle modifications not only lower blood glucose levels in the body, but also, they ameliorate many risk factors for cardiovascular disease, and help weightloss. However, most patients cannot have a good lifestyle [13], so patients must depend on medications for treatment.

The present treatment of DM is focused on controlling and lowering blood glucose levels in the vessel to a normal level [11]. The mechanisms of both modern medicines and traditional medicines to lower blood glucose concentration are: (1) to stimulate beta-cell of the pancreatic islet to release more insulin; (2) inhibit the action of hormones which increases blood glucose concentration; (3) increase the sensitivity of insulin receptor site; (4) inhibit hydrolysis of glycogen in liver; (5) enhance the use of glucose in tissues and organ [14,15,16].

Currently, there are six main classes of modern medicines used all over the world for controlling blood glucose levels and two classes of injections [12]. The tablets are known as biguanides (metformin), sulfonylureas, thiazolidinediones (glitazones), meglitinides (glinides), alpha-glucosidase inhibitors and DPP-4 inhibitors. The classes of medications given by injection are incretin mimetics and insulin [17,18,19,20,21,22,23,24,25,26]. Mechanisms of these medications have been reported. However, most modern drugs have many side effects and adverse effects, causing some serious medical problems during medication processing. Metformin is one medicine of biguanides which can inhibit the production of glucose molecules in the liver of human and increase insulin sensitivity. However, metformin also causes some main side effects such as gastrointestinal on the initial state, including dyspepsia, nausea and diarrhea. Metformin should be avoided in those with severely compromised renal function, de-compensated heart failure, severe liver disease other serious medical problems. The beneficial effects of thiazolidinediones in diabetes treatment are to improve the sensitivity of insulin, decrease insulin resistance and also decrease cardiovascular risks. However, the most common adverse effects of thiazolidinediones are weight gain and fluid retention, leading to peripheral edema and heart failure. The drugs were avoided to use for patients in similar situations, including heart failure and severe liver problems. Rosiglitazone may have cardiovascular risks, and increase the risk of a heart attack. Pioglitazone has not been available in some countries due to concerns about an increased risk for bladder cancer.

According to IDF’s statistics, health expenditure has been increased, increasing from USD 232 billion spent worldwide in 2007, to USD 760 billion in 2019 for adults aged 20–79 years [7]. The economic impact of diabetes has been predicted to continue to increase regularly. The health expenditure can be predicted to reach USD 825 billion by 2030 and USD 845 billion by 2045. Therefore, medication expenditure has become a serious problem for patients with diabetes from developing countries.

Besides modern medication, traditional medicines have been used for a long time and play an important role as alternative medicines [27,28,29,30]. According to WHO, a plant-based traditional system of medicine is still the chief support of about 75–80% of the world population mainly in developing countries having a diversity of plants [31]. Traditional medicines are usually the first choice for primary healthcare of patients in developing countries because of better cultural acceptability, better compatibility with the human body and lesser side effects than modern therapies. Recently, some medicinal plants have been reported to be useful in diabetes worldwide and have been used empirically as antidiabetic and antihyperlipidemic remedies. More than 400 plant species having hypoglycemic activity have been available in the literature [32], however, investigating new antidiabetic drugs from natural plants has still been attractive because they contain phytoconstituents that demonstrate alternative and safe effects on the treatment of diabetes mellitus. Most plants contain bioactive components, such as phenolics, glycosides, alkaloids, terpenoids, flavonoids, carotenoids, etc., that have been improved as having antidiabetic activities [33,34,35,36].

## 3. Bioactive Compounds from Plants Having Type 2 Antidiabetic Activity

The hypoglycemic and antidiabetic effects of several plants used as traditional antidiabetic remedies have been proved, and the mechanisms of hypoglycemic activity of these plants have been studied effectively [35,36,37]. This review focuses on the mechanism of traditional herbal and natural medicines from traditional medicinal plants for diabetes treatment.

### 3.1. Bioactive Compounds Act as Insulin

#### 3.1.1. *Momordica charantia* (Bitter Melon)

*Momordica charantia* (MC) is one of the most common vegetables in the tropical region, particularly in Vietnam, India, China, East Africa, South–North Asia, and Central and South America [38,39]. It is a member of the Cucurbitaceae family and is known as bitter melon or bitter gourd (Figure 3). Besides using MC as a vegetable, it is supposed to be a herbal medicine, used as folk medicine. Its bioactivities, such as anti-inflammatory activity, anti-oxidant activity, anti-viral activity, anti-cancer activity, anti-bacterial activity, etc. and especially anti-diabetic activity [40].

##### Phytochemistry

There were many investigations publishing active components of bitter melon that support type 2 DM treatment. The important phytochemicals of the plants are steroids, momordicosides (A, B, C, D, E, G, F_1_, F_2_, I, K, L), acyl glucosyl sterols, fatty acids, amino acids, alkaloids, phenolic compounds, steroidal saponin, vitamins, carbohydrates, and minerals, etc. [40,41,42,43,44,45,46,47,48].

##### Antidiabetic Activity

The fruits, seeds and callus of *Momordica charantia* contain some insulin-like proteins [49] which are homologous to human insulin, and it produced consistent hypoglycemic effect when tested on rats, gerbils, langurs and human beings [50]. In India and China, MC was believed to be a treatment for diabetes mellitus for thousands of years. Nowadays, scientists have done many types of research focusing on its anti-hyperglycemic abilities. Indeed, many research papers have shown that its bioactivities decrease significantly in blood glucose levels (Figure 4). These investigations on bitter melon also demonstrated that it can enhance the glucose tolerance of normal and diabetic mice and also in humans [47,48,49,50,51]. Many studies proved that bioactive constituents for MC have considerable antidiabetic activities (Table 1).

Akhtar N et al. reported that Charantin, vicine and polypeptide-p contained in ethanolic extract of MC fresh green whole fruit acted as insulin-like protein, decrease blood glucose level, stimulated insulin secretion, tissue glucose uptake, decreased hepatic gluconeogenesis, stimulated glucose uptake and utilization, inhibited absorption of glucose in the intestine in alloxan-induced diabetic rabbits by 200 mg/kg b.w dose [52]. In vitro insulin secretion assay, Momordicoside U (3β,7β-Dihydroxycucurbita-5,23(E)-dien-19-al-25-O-β-d-glucopyranoside) showed that this compound could enhance the uptake of glucose by evaluation of insulin secretion activity [53]. According to Singh N et al., alcoholic extract of the whole fruit which was orally administered in adult alloxan-induced diabetic albino rats can recover β-cells of the islets of Langerhans of the pancreas by 25 mg, 50 mg and 75 mg doses of the extract [54]. In the acetone extract of this fruit power, the ability of recovery of beta cells of the islets of Langerhans of the pancreas was confirmed when applied to alloxan diabetic albino rats by 25 mg, 50 mg and 75 mg doses of extract in 8 to 30 days treatment [55]. An investigation on the antidiabetic activities of MC fruit juice in streptozotocin-induced diabetic rats showed that this herb possesses the ability to enhance insulin secretion of pancreatic β-cells, elevate of serum insulin level, improve β-cell function, decrease insulin resistance, increase glucose utilization in a dose of 10 mL/kg/day for 14 days [56]. In cells of the pancreas, *Momordica charantia* can be able to renew or recover the partially destroyed cells and encourage the secretion of insulin.

Moreover, aqueous ethanolic extract of MC seeds exposed the protection of pancreatic β-cells in vitro experiment [57]. In the treatment of type 1 diabetes mellitus, MC extract has been proven to possibly definitely improve beta cells in pancreatic islets in streptozotocin-induced type-1 diabetic rats. Related with type-1 diabetes, polypeptide-p has shown action similar to human insulin in the body and, therefore, may be used as plant-based insulin replacement in patients with type-1 diabetes [58]. Other studies also investigated the inhibition of diabetes-related enzymes, such as alpha-glucosidase and alpha-amylase from MC extracts [59].

#### 3.1.2. *Panax ginseng* C.A Meyer

In Korea, Ginseng is the most famous traditional plant used in folk medicine for a long time (Figure 5). Ginseng belongs to the genus Panax in family Araliaceae [60]. It distributes typically in a cooler climate region that can be found in Eastern Asia such as Korea, Eastern Siberia, Northeast China and North America.

##### Phytochemistry

The root of this plant contains many bioactive compounds, including triterpene glycosides, or saponins, commonly referred to as ginsenosides (Table 2), Panaxans, vanillic acid, salicylates. All parts of the plant also have some active constituent, such as amino acids, alkaloids, phenols, proteins, polypeptides, and vitamins B1 and B2 which have been identified [61,62,63].

##### Antidiabetic Activity

Since diabetes mellitus is characterized by insulin resistance, and β-cell dysfunction, therapeutic medications should be involved in improving insulin resistance, enhancing glucose uptake, decreasing blood glucose concentration, and protecting/regenerating β-cell from pancreatic islets. Many researchers have been investigated the anti-diabetic activities on the root of *Panax ginseng* in vitro and in vivo experiments [62,63,64]. The most important group of phytochemicals f *Panax ginseng* is ginseng-specific saponins called ginsenosides. Among them, Ginsenosied Rb2 was the most effective constituent treated for streptozotocin-induced diabetic rats by decreasing blood glucose level [65]. Moreover, in fermented red ginseng extracts, the content of ginsenoside Rg2, Rg3, and Rh2 are higher than normal ginseng so that those extracts significantly reduced blood glucose levels and increased plasma insulin levels in streptozotocin-induced diabetic rats by orally-administered 100 or 200 mg/kg extracts dissolved in water, at 10 a.m. daily in three weeks [66]. These mechanisms have been displayed in Figure 6. In general, saponins, which were isolated from ginseng that has been proven significant antidiabetic activity. The mechanism of these components in antidiabetic treatment is to moderate the enzyme activity [50,60] to influence glucose metabolism and control insulin secretion [67,68,69,70,71,72,73].

### 3.2. Bioactive Compounds Increase Insulin Secretion from Beta-Cells of Pancreas

#### 3.2.1. *Allium cepa* (Allium)

*Allium cepa* Linn. is a member of the family Liliaceae which is commonly called onion. It is a useful ingredient in cooking in many countries, especially Vietnam, China and Egypt. This plant can survive under harsh conditions, e.g., winter or dryness, so that it can be stored for a long time without any changes in phytonutrients [74].

##### Phytochemistry

Onion has large numbers of biologically active compounds. These phytoconstituents have been successfully isolated and identified, including phenolics, flavonoids, thiosulfinates, amino acids, essential oils, and vitamins, etc. onion extracts exhibited various bioactivities, such as anti-inflammatory activity, antioxidant activity, and antidiabetic activity, etc. The major biochemical constituents of onion extraction were identified as quercetin, allicin (S-oxodiallyl disulfide), alliin (S-allyl L-cysteine S-oxide), diallyl disulfide (allyl disulfide), S-methyl L-cysteine S-oxide (3-(methyl sulfinyl alanine), propanethial S-oxide (thiopropanal S-oxide) and 3-mercapto-2-methypentan-1-ol [75,76].

##### Antidiabetic Activity

Hypoglycemic activity of *Allium cepa* Linn. extracts have been reported [77]. The bulb part contains S-methyl cysteine sulfoxide, S-allyl cysteine sulfoxide has been proven anti-diabetic activity. These compounds can lower blood glucose levels and has a potent antioxidant activity which may account for hypoglycemic potential. S-methylcysteine sulfoxide (Figure 7) exerts antidiabetic action in 3 different ways: (1) stimulate the production of insulin in the body and enhance the secretion of the pancreas; (2) interfere with dietary glucose absorption; and (3) use insulin effectively [78,79].

#### 3.2.2. *Allium sativum* (Allium)

*Allium sativum* Linn., is commonly herb which has been known as garlic, belongs to the family Allium. It could be found in Asia, Africa and Europe. It is originally indigenous to Asia, is now widely grown in popularity to produce a condiment, especially in Asian cuisine [80].

Their functions prove that garlic is a powerful medicinal plant for treating many diseases for over more than a thousand years. According to numerous scientific research studies, garlic has been discovered as having a wide range of biological functions such as anti-tumor activity, being antibiotics with antimicrobial activity and especially anti-hyperglycemic activity [81].

##### Phytochemistry

Raw garlic contains many active phytochemicals like alkaloids, flavonoids, cardiac glycosides, terpenes and steroids, resin. It also contains some sulfur compounds, such as alliin, allicin, ajoene, diallyl sulfide, enzymes, B-vitamins, proteins, minerals, saponins, flavonoids, etc., which are not Sulphur-containing compounds [81,82,83,84].

##### Antidiabetic Activity

Depending on these scientific studies, garlic’s biological activity in anti-diabetics has been shown that its mechanism is to control the excretion of insulin from β-cells, enhance glucose tolerance and glycogen synthesis [85]. For example, these two bioactive compounds, which are extracted from garlic, are allyl propyldisulfide and S-methylcysteine sulfoxide can decrease blood glucose level. In addition, the ethanol extract from garlic was also had antidiabetic activity by restoring delayed insulin response [86,87].

#### 3.2.3. *Aloe vera* L. Burm. (Asphodelaceae)

*Aloe vera* is the popularly medicinal plant ever known and the most applied medicinal plant especially in the cosmetic industry, and antidiabetic mediation (Figure 8). This traditional medicinal plant belongs to the family Liliaceae. It is original to Africa and Mediterranean countries. It is reported to be distributed widely in the islands of Cyprus, Malta, Sicily, Cape Verde and India [88].

##### Phytochemistry

Phytoconstituents in the plant are alkaloids, flavonoids, tannins, phenols, saponins, carbohydrates, vitamins and minerals and several other aromatic compounds [89]. These compounds have been proven for various pharmacological activities, such as antioxidant, antimicrobial, antidiabetic, anti-cancer and so on. That is the result why until now scientists continue to investigate biological activities of this plant to production modern medicine and traditional medicine.

##### Antidiabetic Activity

The experiment on diabetic rats treated with *Aloe vera* water extract orally led to reducing significant blood glucose levels. Statistical analysis of results found that *Aloe vera* water extract is antidiabetic with fewer side effects [90,91]. Moreover less expensive cost is also a significant benefit of *Aloe vera* in the production of medicine against diabetes mellitus.

### 3.3. Bioactive Compounds Regenerate of Beta-Cells of the Islets of the Pancreas

#### 3.3.1. *Pterocarpus marsupium* (Fabaceae)

*Pterocarpus marsupium* (*P. marsupium*) is a large, high tree that can grow up from 15 to 30 m tall (Figure 9). It belongs to the Fabaceae family and distributes in India, Nepal, and Sri Lanka, which is widely used in ‘Ayurveda’ as ‘Rasayana’ for the management of various metabolic disorders including hyperglycemia [92].

##### Phytochemistry

Similar to most herbs, *P. marsupium* is a rich source of phenolic and flavonoid compounds. It has been reported that this plant contains alkaloids, steroids, terpenoids, tannins, amino acids, proteins, etc. The potential antidiabetic constituents in this plant have been identified successfully, including epicatechin. It is also reported to be rich in polyphenolic compounds which have been considered major bioactive compounds, such as marsupsin, pterosupin and pterostilbene and flavonoids pteroside, pteroisoauroside, carsupin and marsupol. These compounds have many biological effects, e.g., anti-inflammatory, anti-bacterial, anti-oxidant activity, especially antidiabetic activity [93,94,95,96].

##### Antidiabetic Activity

The extract of *P. marsupium* was orally administered by diabetes patients and gave a strong anti-hyperglycemic effect [92,94]. Ethyl acetate’s extract of *P. marsupium* was tested for five days on alloxan-induced diabetic rats to conclude the significant anti-diabetes effect. The ethanolic extract of *P. marsupium* showed the blood sugar lowering effect [95]. It was found to be effective in lowering the glucose level. Epicatechin, isolated from *P. marsupium*, showed the ability of regeneration of the β-cells of the pancreas islets [97]. Moreover, the aqueous extract of this plant was reported that it could stimulate the secretion of insulin and enhance the glucose uptake, so that it can be considered as antidiabetic medicine [94].

#### 3.3.2. *Tinospora cordifolia* (Menispermaceae)

*Tinospora cordifolia* (*T. cordifolia*) has been commonly known as “Amrita” or “Guduchi”, belong to the Menispermaceae family. It has been one of the important drugs of Indian Systems of Medicine for a long time. Guduchi is native to India and mainly distributed in tropical areas such as Myanmar and Sri Lanka [98]. In India, this plant has been reported as the main source of treatments for many diseases such as fever, dyspepsia and urinary diseases in folk medicine [99].

##### Phytochemistry

*Tinospora cordifolia* has been reported to contain numerous constituents belonging to different chemical classes of secondary metabolites such as alkaloids, terpenoids, essential oils, glycosides, steroids, phenolic constituents, aliphatic compounds, and polysaccharides. Leaves of this plant are a rich source of proteins, flavonoids, alkaloids and glycosides [100,101]. These active compounds have been exposed to several biological activities, including antiseptic, anti-inflammatory, anti-cancer, antimicrobial and antidiabetic activities.

##### Antidiabetic Activity

*T. cordifolia*, containing polysaccharide isolated from this plant exposed the β-cell regenerative properties which could be pointed to develop antidiabetic medicine with few side effects [102]. Oral administration of the extract of *T. cordifolia* roots for two weeks experimented with induced type 2 diabetic rats resulted in this plant can promote insulin secretion and inhibit glucosgenolysis process and therefore improve the regulation of blood glucose level in the body [103]. In 2011, Patel also studied the hypoglycemic effects of alkaloidal fraction of *T. cordifolia* extract. This investigation demonstrated the antidiabetic activity of *T. cordifolia* due to the promoting of insulin-releasing, improving insulin sensitivity and inhibiting gluconeogenesis process [104].

#### 3.3.3. *Tinospora crispa* (Menispermaceae)

*Tinospora crispa* (*T. crispa*) is a precious medicinal climbing plant. As a kind of vines, its body is very rough, light brown, long to 6–7 m or more (Figure 10). It wildly grows in many South East Asia countries such as Laos, Cambodia, Thailand, and Philippines, etc., particularly common in the Northern areas in Vietnam [105,106,107].

##### Phytochemistry

In many previous investigations, *T. crispa* is a rich source of secondary metabolites, divided into several groups, including alkaloids, flavonoid, terpenoids, lignans, sterols, etc. [105]. Alkaloidal constituents have been isolated including *N*-formylasimilobine 2-O-β-d-glucopyranoside, *N*-formylasimilobine 2-O-β-d-glucopyranosyl-(1→2)-β-d-glucopyranoside, (tinoscorside A), *N*-formylanonaine, *N*-formyldehydroanonaine, *N*-formylnomuciferine, *N*-demethyl-*N*-formyldehydronornuciferine, magnoflorine, etc. [105,106,107,108]. Until now, there are several flavones and flavone glycosides which have been isolated and identified from *T. crispa* extract, such as apeginin, diosmetin, genkwanin, luteolin 4′-methyl ether 7-glucoside, genkwanin 7-glucoside, luteolin 4′-methyl ether 3′-glucoside [109]. Moreover, many researchers reported that this plant also contains a lot of phytochemical bioactive constituents [110].

##### Antidiabetic Activity

The investigation of Noor and Ashcroft [106] indicated that the orally administrated extract of *T. crispa* exhibited a potential antidiabetic effect. The mechanisms of these activities were predicted so that this plant could stimulate insulin secretion through the modulation of β-cell Ca^2+^ concentration. In Noipha’s experiment, *T. crispa* extracts improved the glucose transport activity of L6 myotubes by increasing GLUT1 transporter [110]. Thus, it can be further used as an antidiabetic agent for the treatment of type II diabetes [110,111].

#### 3.3.4. *Gymnema sylvestre* (Apocynaceae)

*Gymnema sylvestre* (*G. sylvestre*) belonging to the Apocynaceae family, originated from tropical forests of The Southern and Central India and Sri Lanka [112]. In Vietnam, *G. sylvestre* has been classified as vines and found in many Northern and Central provinces. So far, this plant has been widely used in several countries around the world in treating diabetes.

##### Phytochemistry

The phytoconstituents of *G. sylvestre* consist of many groups containing bioactive compounds that are listed in the below table (Table 3) [112,113].

##### Antidiabetic Activity

The major biological active ingredient of *G. sylvestre* is Gymnemic acids—a group of triterpenoid saponins isolated and identified successfully (Figure 11). Several studies have reported that the main biological activity of this plant is antidiabetic activity [113].

In the investigation of S. Sathya et al., the aqueous extract of *G. sylvestre* leaf informed hypoglycemic activity in normal and alloxan-induced diabetic rats [114] by reducing glucose concentration. Other research on *G. sylvestre* extracts suggested that this extract could stimulate the release of insulin in vitro by permeabilizing the β-cell [115]. Several mechanisms have been proposed to explain the antidiabetic activity of *G. sylvestre* (Figure 12) [113,116,117]. According to these literature, the action of Gymnemic acids in diabetic treatment was reported to be able to stimulate pancreatic cell production, thereby increasing insulin production, increase insulin sensitivity and insulin activity, help to control and stabilize blood glucose concentration in the body. Gymnemic acids were also reported to be able to inhibit the absorption of glucose in the small intestine and inhibit the conversion glycogen in live to glucose molecules in blood [118].

### 3.4. Bioactive Compounds Reduce the Absorption of Glucose from Gastrointestinal Tract

#### 3.4.1. *Cyamposis tertragonoloba* (Fabaceae)

*Cyamopsis tetragonoloba* L. Taub is commonly known as Cluster Bean or Guar, belongs to the Fabaceae family. The plant is a very small drought-resistant herb and widely cultivated in India. This plant was reported to contains a high amount of carbohydrates and fiber [119].

##### Phytochemistry

There are very few phytochemical studies carried out on this plant. Some phytoconstituents in leaves were reported to contain carbohydrates, proteins, fibers, galactomannans, ascorbic acid and condensed tannins together with, caffic acid, gallic acid, and gentisic acid, *p*-coumaric acid, astragalin, *p*-hydroxycinamyl and coniferyl alcohol. Its flavonoid content includes quercetin, daidizein, kaemferol, and its 3-arabinosides. Moreover, other polyphenolic compositions of the plant were identified, including gallotannins, gallic acid and gallic acid derivatives, myricetin-7-glucoside-3-glycoside, chlorogenic acid, ellegic acid, 2,4,3-trihydroxy benzoic acid, texasin-7-O-glucoside and *p*-coumaryl quinic acid [119].

##### Antidiabetic Activity

Because of the presence of flavonoids and other phenolic compounds of the plant, *Cyamopsis tetragonoloba* showed a marginal antihyperglycemic effect on blood glucose level in normal fasted rats, however, the blood glucose-lowering effect was significant in alloxan-induced hyperglycemic rats. Other studies showed that this herb could improve insulin release and decrease the amount of HbA1c [120,121]. Gopalsamy Rajiv Gandhi et al. reported that polyphenols identified in *C. tetragonoloba* beans showed hypoglycemic action and protected β-cells [122]. Hence this plant can be considered for use in the management of type 2 diabetes mellitus.

#### 3.4.2. *Ocimum sanctum* L. (Lamiaceae)

*Ocimum sanctum* L. is a small tree, up to 1 m tall, commonly known as ‘Sacred basil’ or ‘Holy basil’ or “Hương Nhu Tía” in Vietnam, is grown as a household plant in many countries for medicinal purposes like Vietnam and India (Figure 13). It belongs to the Lamiaceae family. The whole plant has a pleasant aroma [123,124].

##### Phytochemistry

The phytochemistry of *Ocimum sanctum* is identified in all parts of this plant, containing many nutrients and bioactive constituents. However, the quantity of these constituents depends on many natural factors, including growing, harvesting, storage conditions [125,126].

In leaf extract of *O. sanctum*, volatile oil was extracted and identified chemical compositions, containing many major components like eugenol, methyleugenol, and *p*-caryophylen. The essential oil of this herb contains various bioactive compounds, such as terpenoids, esters, aliphatic aldehydes and phenolic acids. This herb also consists of a diversity of second metabolites, including phenolics, flavonoids, terpenoids, lignans, steroids, fatty acids and their derivatives. These components have been mainly studied for therapeutic purposes because of their biological and pharmacological effects, including antioxidant, anti-inflammatory, antimicrobial, anticancer and antidiabetic activity [125].

##### Antidiabetic Activity

In Hannan’s study, there are four fractions of *Ocimum sanctum* including the ethanol, aqueous, butanol, and ethyl acetate fractions prepared to elucidate the mechanism of antihyperglycemic activity of this plant showed in literature [127]. This result proved that these fractions could stimulate insulin secretion. This study also indicates that the ethanol extract could decrease blood glucose concentration and increase insulin secretion, thereby this plant is a potential herb in diabetic treatment. Moreover, by in vivo experiment, the *O. sanctum* extract was showed to be able to improve oral glucose tolerance, decrease serum glucose, increase glycogen synthesis in the liver [128].

It was reported that leaf power extract lowered plasma glucose level by the presence of many active phytochemicals including eugenol, carvacrol, linalool, caryophylline, β-sitosterol which have been studied about potent hypoglycemic effects efficiency [76]. The other investigation has reported the antidiabetic and hypoglycemic activities of a triterpenoid (16-hydroxy-4,4,10,13-tetramethyl-17-(4-methyl-pentyl)-hexadecahydro-cyclopenta[a]phenanthrene-3-one) isolated from *Ocimum sanctum* by in vivo investigation [129]. The mechanisms of the antidiabetic and hypoglycemic potential of this compound were elucidated to increase the pancreatic secretion of insulin from β-cells, and enhance glucose utilization [129]. It was suggested that this triterpenoid should be considered to be developed as a potential antidiabetic medicine. This evidences support that *O. sanctum* has many benefits in the management of diabetes, and this plant should be encouraged to be a potential anti-diabetic activity [130].

### 3.5. Bioactive Compounds Have Oxygen Radical Scavenging Activity

As displayed above, the fruit juice of *Momordica charantia* herb was determined to be an antioxidant composition because of the radical scavenging effect of the butanol fraction from *M. charantia* extract [131]. As the same result, in 2014, Tsai TH et al. demonstrated antioxidant activity and cell-protective activity of *M. charantia* extract because this herb extract could scavenge several free radicals [132]. The antidiabetic action of *Momordica charantia* has been elucidated through this mechanism. The efficacy of some Indian ayurvedic plant extracts, including *Momordica charantia*, *Eugenia jambolana* (Figure 14), *Tinospora cordifolia* and *Mucuna pruriens* (Figure 15) was investigated for the prevention of diabetic cataract in vivo. The results showed that *Momordica charantia* and *Eugenia jambolana* had better protective activity in the development of diabetic cataract as compared to *Tinospora cordifolia* and *Mucuna pruriens* [133].

### 3.6. Bioactive Compounds Inhibit Alpha-Glucosidase and Alpha-Amylase Activities

Salivary or pancreatic alpha-amylase is an enzyme that catalyzes to hydrolyze α-bonds of large, α-linked polysaccharides, such as starch in plants and glycogen in animals and humans, to create monosaccharide as glucose. α-glucosidase is an enzyme located in the small intestine that hydrolyses terminal non-reducing (1→4)-linked α-glucose residues to release a single α-glucose molecule. α-amylase and α-glucosidase are important digestive enzymes affecting blood glucose concentration, especially with patients in diabetes mellitus. The action of these enzymes increases glucose concentration causing hyperlycemic status related to diabetes mellitus. So, the inhibition of the action of these enzymes is one of the most important keys in diabetes treatment [134,135,136,137,138]. Many herbs have been reported so that various phytoconstituents isolated from these plants had potential inhibitory activities of two enzymes, thereby focusing on traditional medicine has been noticed.

#### 3.6.1. *Costus pictus* (Zingiberacea)

*Costus pictus* D. Don, commonly known as ‘insulin plant’ belongs to the Zingiberacea family (Figure 16). It has been famous for antidiabetic activity and is used as a traditional dietary supplement for diabetes treatment in Southern India [139].

##### Phytochemistry

The analysis of phytochemicals of *C. pictus* has been reported that most parts of insulin plant including stem, leaves, rhizome and flowers are a diversity of primary and secondary metabolites [139]. Among them, there are many groups of secondary metabolites, including alkaloids, saccharides, terpenoids, glycosides, steroids, tannins, saponins, phenols, flavonoids, etc. [140,141,142,143]. Besides the extract of this herb also contains trace elements such as K, Ca, Cr, Mn, Cu, and Zn [144].

##### Antidiabetic Activity

Until now, various investigations have been carried on an insulin plant to determine anti-diabetic activity [143]. The in vitro study was undertaken to investigate the effects of the *C. pictus*. The result of that investigation showed that the significant mechanism of antidiabetic activity of *Costus pictus* is the α-amylase and α-glucosidase inhibitory activity. In other research, the ethanolic and methanolic extracts of *C. pictus* which have been administrated orally to streptozotocin and alloxan-induced rats have shown a significant reduction in the blood sugar level, and also could increase insulin level [144]. Along with the identification of trace elements like K, Ca, Cr, Mn, Cu, and Zn stimulates the antidiabetic property. The other study also reported that antidiabetic property on streptozotocin-induced diabetic rats and in vitro pancreatic islet culture of *C. pictus*. Inhibition of carbohydrates hydrolysing enzymes activity like α-amylase and α-glucosidase also gave good results [145,146]. In 2009, G. Gireesh et al. carried out an in vivo experiment of *C. pistus* leaf to determine antihyperglycemic activity and insulin secretion in diabetic rats. Their results suggested that insulin plant could lower glucose concentration, increase insulin secretion, and also enhance glucose utilization [147]. Another in vitro study in 2010 showed that *Costus pictus* extracts could stimulate insulin secretion from β-cells of islets of Langerhans [148]. 

#### 3.6.2. *Phaseolus vulgaris* (Leguminosae)

*Phaseolus vulgaris* L. (*P. vulgaris*) belongs to the Leguminosae family, commonly known as kidney bean, which is a food item of mass consumption in Asia and Eastern countries [149]. *P. vulgaris* is an annual, herbaceous plant which its main root grows deeply so that this plant has good tolerance with changes of climate. In Vietnam, its fresh beans have been used as a daily vegetable.

##### Phytochemistry

*P. vulgaris* has been reported that is an important source of primary and secondary metabolites necessary in a healthy diet of humans. By using the HPLC technique to separate constituents’ efficiency, the phytochemicals of *P. vulgaris* were identified successfully [149,150,151]. These compounds could be divided into different groups displayed in Table 4.

##### Antidiabetic Activity

Various previous studies have shown that seeds of kidney bean have α-amylase inhibitors inhibiting α-amylase activities in animals and insects, related to control type-2 diabetes mellitus by inhibition of DPP-IV, and a lectin called phytohaemoagglutinin could adjust the activity of glucagon-like peptides GLP-1 (glucagon-like peptide-1) [152,153,154]. The common bean *P. vulgaris* have three isoforms of α-amylase inhibitor including α-AI1 (known as phaseolamin), α-AI2, and α-AIL, which has been identified and tested in numerous clinical studies against α-amylase action [152,155]. The mechanism of action of these enzyme inhibitors indicated that they could reduce the carbohydrate absorption by inhibiting the activity of α-amylases of mammals and insects [155,156]. The reduction of the glycemic index may also limit the risks of insulin resistance in diabetes mellitus, thereby control the serious consequences of the disease. Recently, another investigation continues to determine α-amylase inhibition of *P. vulgaris* extract [157]. The present study also indicated that α-amylase inhibitor inhibited the α-amylase activity significantly, and prolonged treatment with *P. vulgaris* extract could reduce several problems related to diabetes mellitus.

#### 3.6.3. *Euphorbia hirta* Linn. (Euphorbiaceae)

*Euphorbia hirta* Linn. (E. hirta) is a common herb that belongs to *Euphorbia* genus of Euphorbiaceae family. *Euphorbia hirta* L. is found in pan-tropic, partly sub-tropic areas and worldwide including Australia, Western Australia, Northern Australia, Queensland, New south wales, Central America, Africa, Indonesia, Malaysia, Philippines, China and India [158]. In Vietnam, *Euphorbia hirta* L. is commonly distributed in many provinces of the southern area (Figure 17). *Euphorbia hirta* L. was used as a traditional medicine in the treatment of diabetes for a long time ago [158,159,160,161].

The strong anti-diabetes activity of Euphorbiaceae family in general and *E. hirta*, in particular, was reviewed in 2006 [84]. Unambiguously, *E. hirta* can serve as a potential candidate to develop newline anti-diabetes drugs in the future.

##### Phytochemistry

*Euphorbia hirta* contains various bioactive compounds, including flavonoids, terpenoids, phenols, essential oil and other compounds (Table 5). Some flavonoids have been identified in *E. hirta* including quercetin, quercitrin, quercitol and derivatives. Isolated terpenoids in *E. hirta* include triterpenoids, α-amyrin, β-amyrin, friedelin, teraxerol, cycloartenol, 24-methylene-cycloartenol, ingenol triacetate β-sitosterol, campestrol, stigmasterol and so on. Some complex tannins have been also determined in *E. hirta* extract, such asdimeric hydrolysable dehydro ellagic tannins, euphorbins A, B, C, E, etc. terchebin, the monomeric hydrolysable tannins geraniin, 2,4,6-tri-o-galloyl-β-d-glucose and 1,2,3,4,6-penta-O-galloyl-β-d-glucose and the esters 5-O-caffeoyl quinic acid (neo chlorogenic acid), 3,4 –di-o-galloyl quinic acid and benzyl gallate [159,160].

##### Anti-Diabetic Activity

The ethanolic extract of *Euphorbia hirta* was reported to be able to decrease blood glucose level (hypoglycemic effect) on alloxan-induced diabetic rats. From other in vitro experiments, ethanol extract and ethyl acetate fractions inhibit α-glucosidase activity. In vivo tests also had the same result. [158,159,160]. The presence of various bioactive compounds, especially in chloroform and ethyl acetate extracts demonstrated the ability in the treatment of diabetes by inhibition of enzymes involving in the metabolism of saccharides such as α-amylase [161]. Polyphenolic compounds have popularly reported inhibitory activity against α-amylase in both in vitro and in vivo experiments. Phenolics and flavonoids found in *E. hirta* such as quercetin, quercitrin, and rutin have been proved to be effective inhibitors of mammalian α-amylase. Although the α-amylase inhibitory activity of *E. hirta* powder is not strong enough as acarbose—a standard drug in diabetes treatment—it also proved that *E. hirta* had mild α-amylase inhibition. Until 2010 Sunil Kumar, Rashmi and D. Kumar presented the evaluation of the antidiabetic activity of *Euphorbia hirta* Linn. in streptozotocin-induced diabetic mice [160]. In 2014, the ethanolic extract of *Euphorbia hirta* was reported that it can decrease blood glucose level (hypoglycemic effect) on alloxan-induced diabetic rats. From other in vitro experiments, ethanol extract and ethyl acetate fractions inhibited α-glucosidase activity. In vivo tests also had the same result [158]. In 2015 Manjur Ali Sheliya and his partners studied the inhibition of α-glucosidase by new prenylated flavonoids from *E. hirta* [162] isolated from ethyl acetate fraction of methanolic extract. By using medium pressure liquid chromatography technique, they successfully isolated four active compounds, including quercetrin (1), dimethoxy quercetrin (2), hirtacoumaroflavonoside (7-O-(*p*-coumaroyl)-5,7,4′-trihydroxy-6-(3,3-dimethyl allyl)-flavonol-3-O-β-d-glucopyranosyl-(2″-1″′)-O-α-l-rhamnopyranoside) (3) and hirtaflavonoside-B (5,7,3′,4′-trihydroxy-6-(3,3-dimethyl allyl)-8-(iso-butenyl)-flavonol-3-C-β-d-glucopyranoside) (4) respectively. The α-glucosidase inhibitory activity of isolated compounds was evaluated and compared with standard drug acarbose. The result showed that those flavonoids increased α-glucosidase inhibition. That study provides credence to the ethnomedicinal use of *E. hirta* in the management of diabetes in folk medicine. In 2016 Manjur Ali Sheliya and his group continued to investigate in vitro α-glucosidase and α-amylase inhibition by aqueous, hydroalcoholic, and alcoholic extract of *E. hirta* [163]. According to their study, the result clearly indicated that the metholic extract of *E. hirta* had strong inhibitory activity against α-glucosidase and quietly mild inhibitory activity against α-amylase, two proteins involved in diabetes mellitus directly. *E. hirta* extract can be used for control of postprandial hyperglycemia with lesser side effects, and provide good material for further studies on treatment type 2 diabetes mellitus.

An in silico study, In 2013 Dr. Ly Le and her group investigated antidiabetic activities of bioactive compounds in *Euphorbia hirta* Linn using molecular docking and pharmacophore [164]. This study demonstrated the effect of bioactive compounds isolated in *E. hirta* on some proteins relating to diabetes type 2, including α-glucosidase and α-amylase by evaluating whether a relationship exists between various bioactive compounds in *E. hirta* and targeted protein relating diabetes type 2 in human. By calculating the binding energy and pharmacophore modeling, eight promising compounds in *E. hirta* were obtained, including cyanidi 3,5-O-diglucose, myricitrin, pelargonium-3,5-diglucose, quercitrin, rutin, α-amyrine, β-amyrine, and taraxerol. The results of that investigation showed that those bioactive compounds have the potential in developing medication for type 2 diabetes mellitus. However, the combination of using the molecular dynamic technique is required to determine more accurate binding affinities and the stability of ligand–proteins’ interactions. Thus, the research results in silico and in vitro studies have shown that this plant has the ability to inhibit two enzymes.

### 3.7. Bioactive Compounds Increase Glucose Utilization

Some medicinal plants such as *Cyamospsis tetragonolobus* (Gowar plant) [165], *Grewia asiatica* (phalsa) [166] and *Zingiber officinale* Rosc (ginger) [166], were reported their hypoglycemic activity by moderating glucose utilization in the body [167]. This part only displays about *Zingiber officinale* Rosc. because this herb has been popular and useful in Vietnam.

#### 3.7.1. *Zingiber officinale* Rosc (Ginger)

It is a traditionally flowering spicy plant in the family Zingiberaceae which was originally native to southern China, and has been grown in many countries in the tropical and subtropical areas, from East Asia to Southeast Asia and South Asia. All parts of this plant, including rhizome, ginger root are widely used as essential food spices or traditional medicine (Figure 18) [168,169].

##### Phytochemistry

The phytochemicals of ginger are quite different depending on the origin and the fresh or dry state of parts of this herb. The phytochemicals of rhizome ginger contain strong free-radical reducing efficacy. They include volatile oils, phenolic compounds and others. Among them, volatile oils, also known as ginger essential oils, are a mixture of terpenoid compounds, including sesquiterpene hydrocarbons, monoterpene hydrocarbons, carbonyl compounds, alcoholsand esters. Especially, the phenolics in ginger are the most important components. The phenolic constituents were divided into two groups: gingerol-, gingeron- and shogaol-related group and diarylheptanoids. Gingerol which is the spicy component of this plant contains a diversity of various bioactive substances. Besides, this plant also contains a variety of amino acids, including glutamate, aspartic acid, serine, glycine, threonine, alanine, etc. Moreover, ginger also contains polysaccharides and organic acids, such as oxalic acid, tartaric acid, etc. [168,169,170].

##### Antidiabetic Activity

Sharma and Shukla reported that ginger juice can lower blood glucose concentration in normal fasting animals and in alloxan diabetic animals [171]. The mechanism of lowering the glucose effect was explained because it can increase the viscosity of gastrointestinal contents, slow gastric emptying and also acts as a barrier to diffusion. Other studies also demonstrated that folk medicinal plant ginger can control tissue glycogen content in diabetic rats by improving the peripheral utilization of glucose and repairing the impaired liver [172].

Moreover, the rhizome of *Zingiber officinale* Rosc also proved that its bioactive components can enhance glucose uptake in cultured L6 myotubes [173]. That investigation suggested that the phenolic gingerol constituents were the major active compounds enhancing glucose uptake. Other investigations showed that the solution of the fresh ginger sample exhibited inhibition against *alpha*-amylase and *alpha*-glucosidase activities and inhibit angiotensin-converting enzyme [174,175].

Furthermore, powder of ginger can decrease the level of glucose and activate inflammatory activity which can lead to the development of insulin resistance [176].

## 4. Conclusions

Diabetes mellitus has been considered to be a major cause affecting the economy of patients, their families and society. Furthermore, uncontrolled diabetes leads to serious chronic complications such as blindness, kidney failure, and heart failure. In order to decrease this problem, researches on new antidiabetic agents are concerned. Because of the adverse effects of modern therapies, many traditional medicines have been noticed. Moreover, herbal extracts nowadays can be used with standard drugs for combinatorial therapies. Each herb has its own active ingredients that can lower blood sugar levels as well as control the complications of diabetes. Future research will focus on isolation, purification, and identification of bioactive substances in plants. This review is looking forward to providing the necessary information in the management of diabetes. In our review, we have introduced a complete list of anti-diabetic plants taken from the Vietherb database [177]. Isolation and identification of bioactive phytochemicals from these plants play an important role in improving insights into anti-diabetic functional food [161] and drug development [178].

## Figures and Tables

**Figure 1 biology-09-00252-f001:**
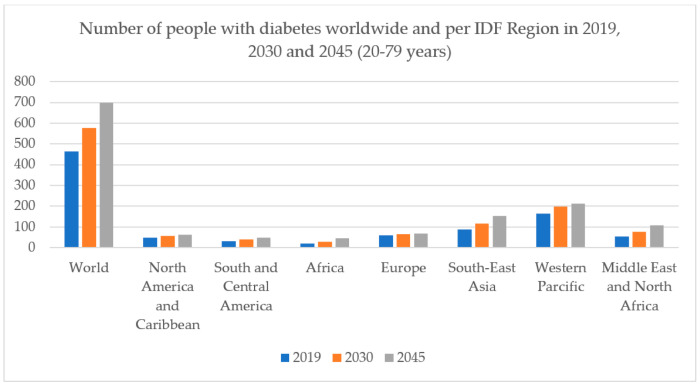
Number of people with diabetes worldwide and per International Diabetes Federation (IDF) region.

**Figure 2 biology-09-00252-f002:**
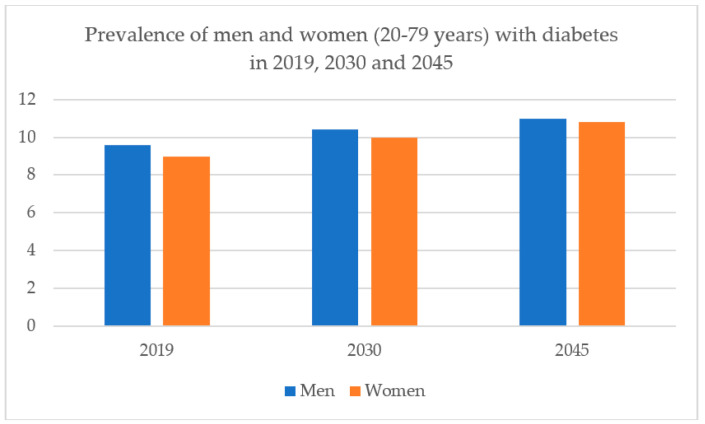
Prevalence of men and women (20–79 years) with diabetes in 2019, 2030 and 2045.

**Figure 3 biology-09-00252-f003:**
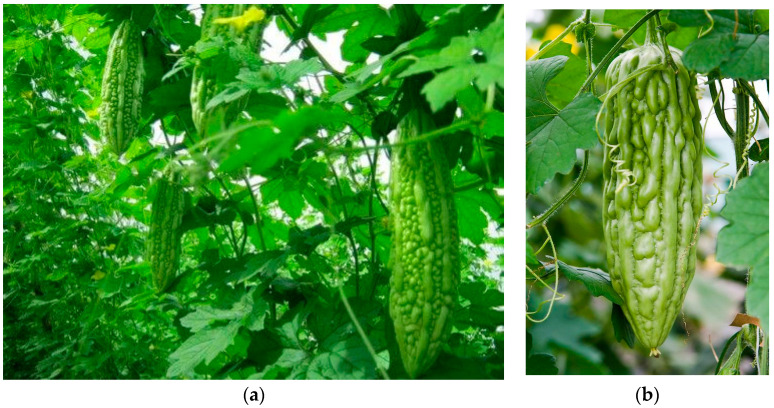
Pictures show the morphological characteristics of the *Momordica charantia* (MC): (**a**) whole plant and (**b**) unripe fruit.

**Figure 4 biology-09-00252-f004:**
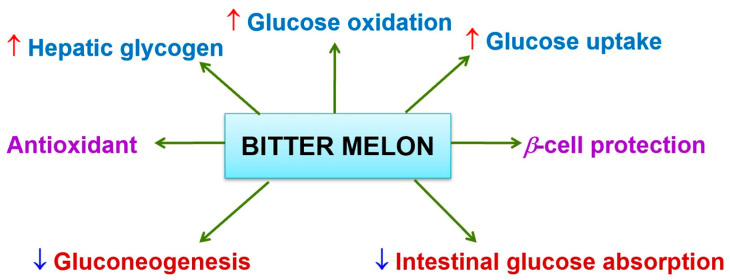
The mechanism in decreasing blood glucose levels of *M. charantia.*

**Figure 5 biology-09-00252-f005:**
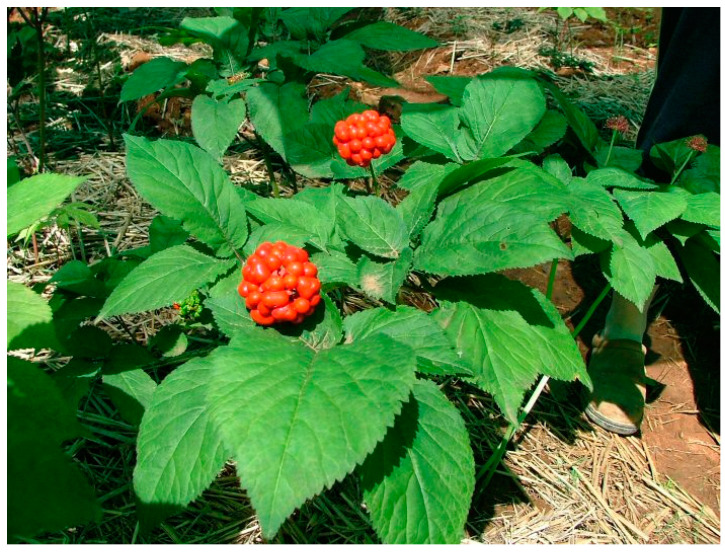
*Panax ginseng* C.A Meyer.

**Figure 6 biology-09-00252-f006:**
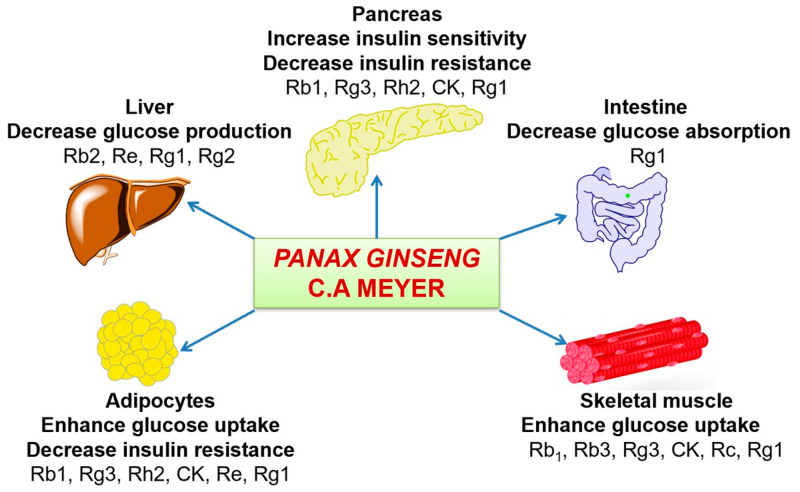
Mechanisms of *Panax ginseng* saponins on different organs related to diabetes.

**Figure 7 biology-09-00252-f007:**
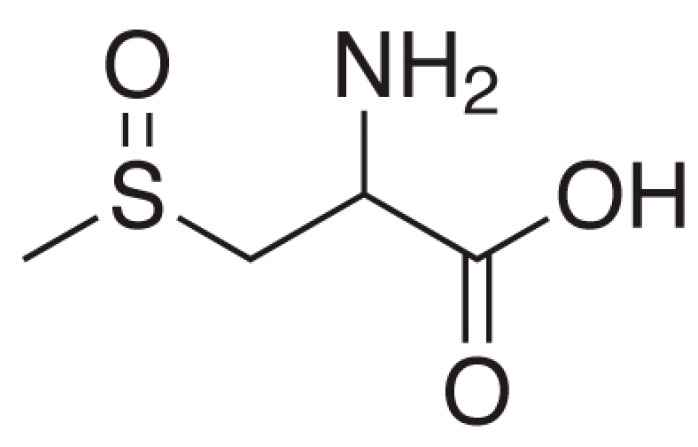
Structure of S-methyl cysteine sulfoxide.

**Figure 8 biology-09-00252-f008:**
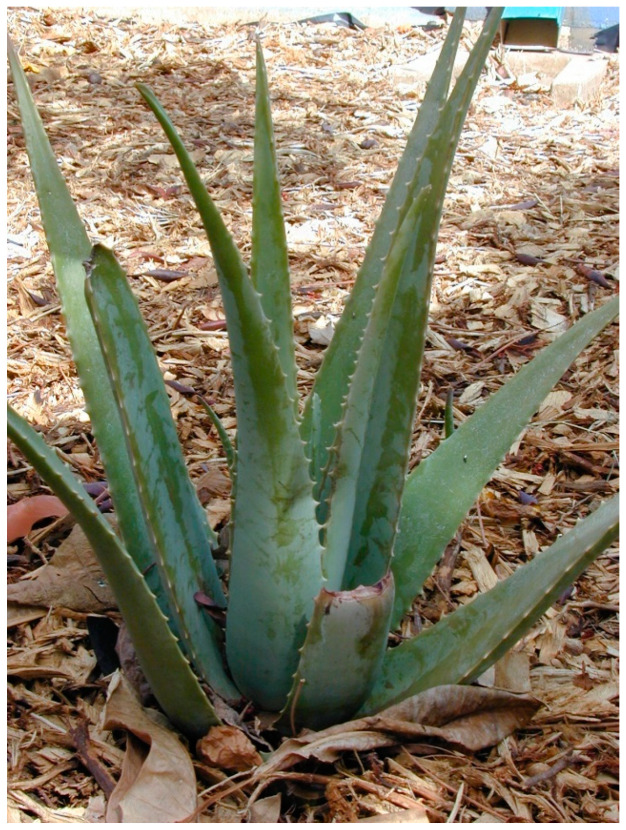
*Aloe vera* L. Burm.

**Figure 9 biology-09-00252-f009:**
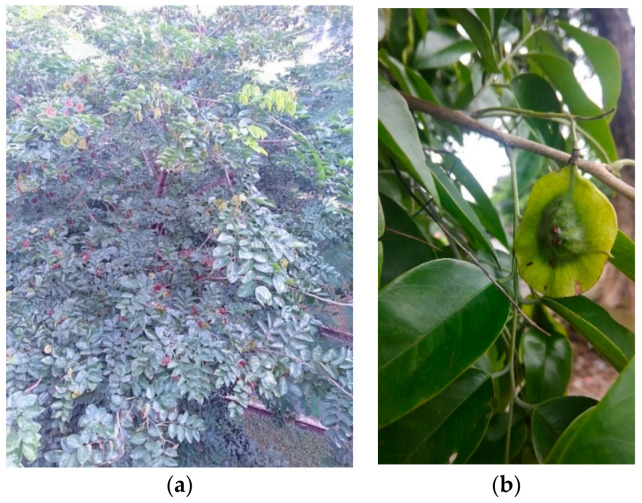
*Pterocarpus marsupium*: (**a**) whole plant and (**b**) leaves.

**Figure 10 biology-09-00252-f010:**
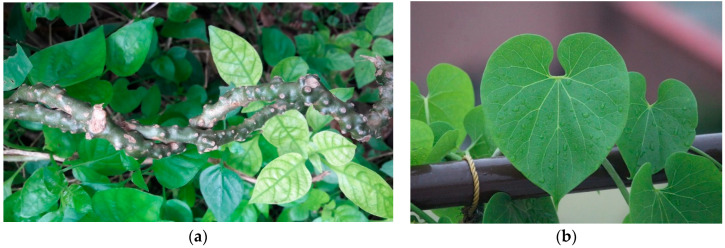
*Tinospora crispa*: (**a**) whole plant and (**b**) leaves.

**Figure 11 biology-09-00252-f011:**
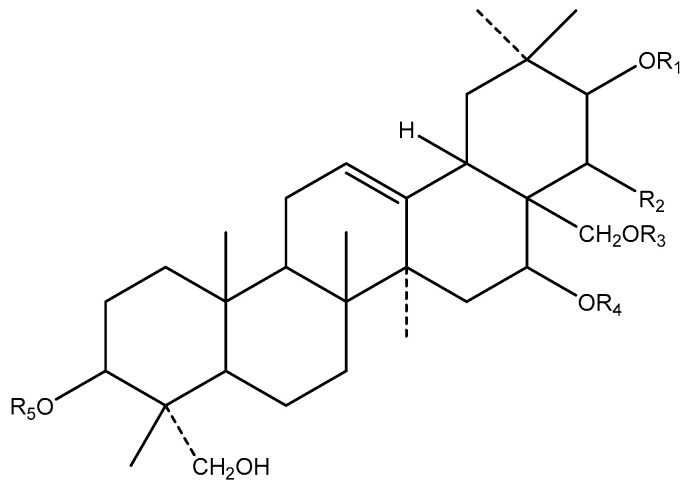
Basic molecular structure of Gymnemic acid.

**Figure 12 biology-09-00252-f012:**
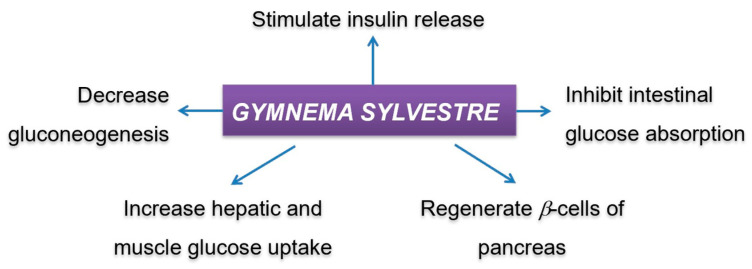
Mechanisms of *Gymnema sylvestre* in the antidiabetic activity.

**Figure 13 biology-09-00252-f013:**
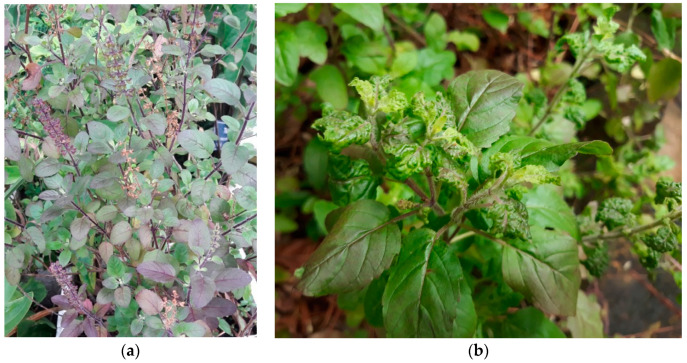
*Ocimum sanctum* L.: (**a**) whole plant and (**b**) leaves.

**Figure 14 biology-09-00252-f014:**
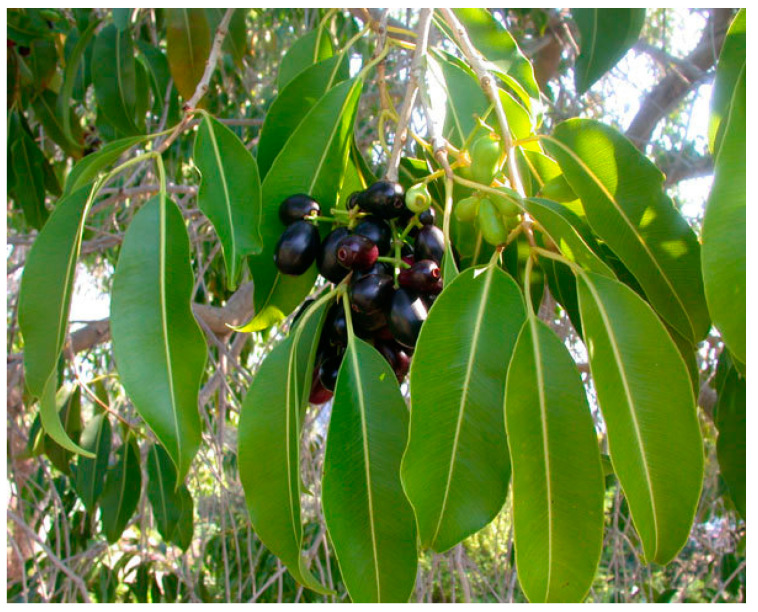
Eugenia jambolana.

**Figure 15 biology-09-00252-f015:**
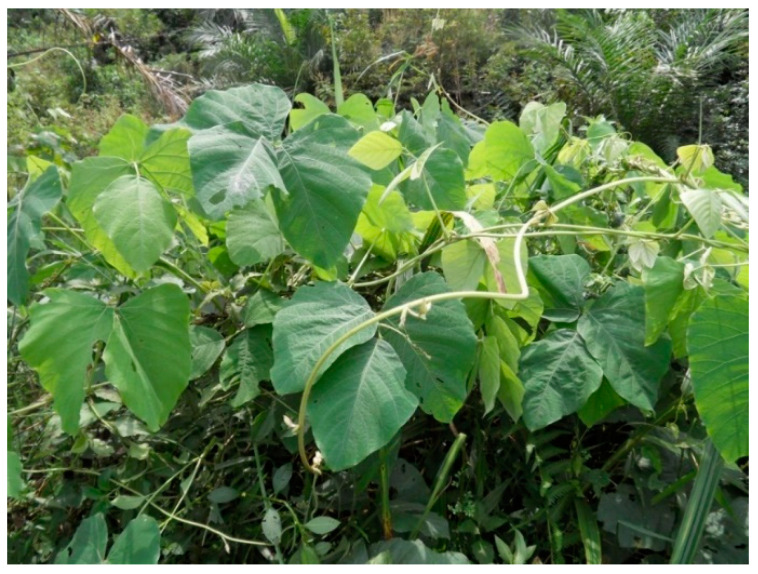
Mucuna pruriens.

**Figure 16 biology-09-00252-f016:**
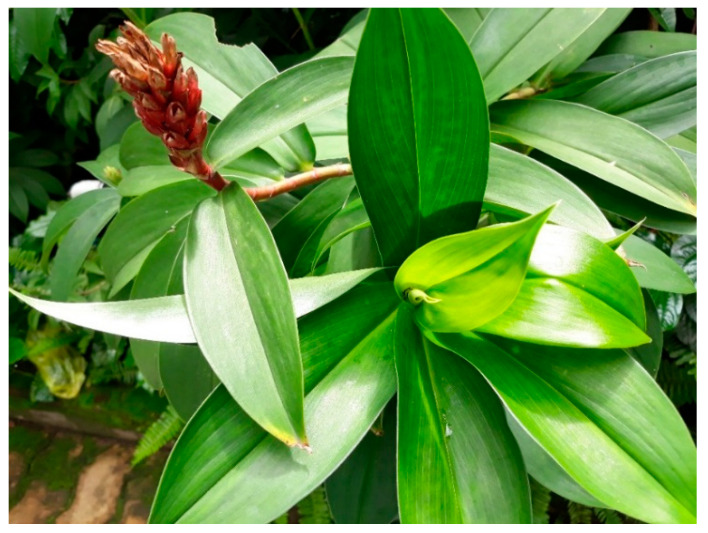
*Costus pictus* D. Don.

**Figure 17 biology-09-00252-f017:**
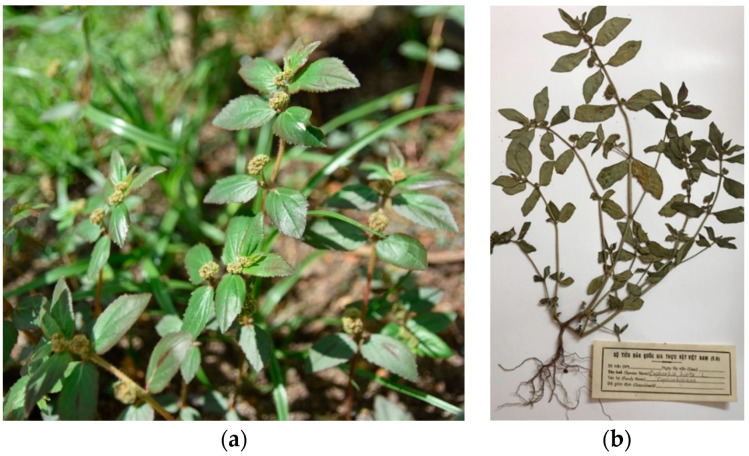
*Euphorbia hirta* Linn.: (**a**) fresh whole plant and (**b**) plant specimen.

**Figure 18 biology-09-00252-f018:**
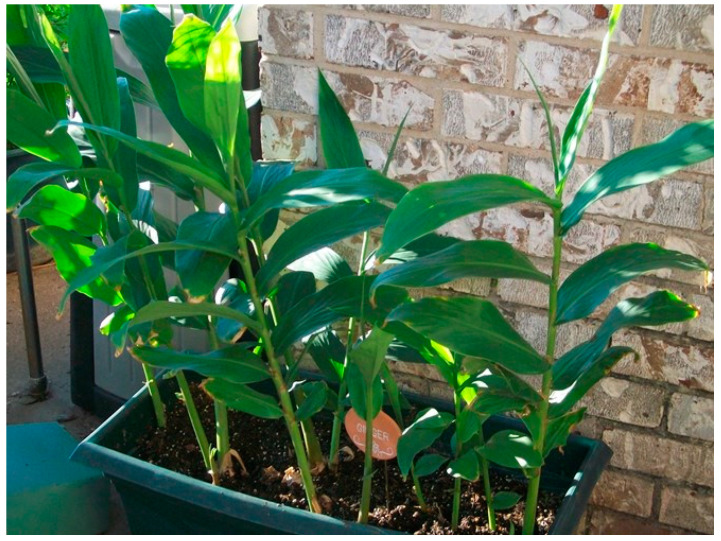
Zingiber officinale Rosc.

**Table 1 biology-09-00252-t001:** Bioactive compounds and their effects in *Momordica charantia*.

Bioactive Compounds	Antidiabetic Effects
Polypeptide-p	Act as Insulin-like protein, decrease blood glucose level
Momordicosides	Enhance the uptake of glucose
Saponins	Stimulate insulin secretion, a lower blood glucose level
Conjugated linolenic acid	Release intestinal GLP-1
Momordin	PPAR δ activation
9c, 11t, 13t conjugated linolenic acid	PPAR α activation

**Table 2 biology-09-00252-t002:** Structure of ginsenosides (ginseng-specific saponins).

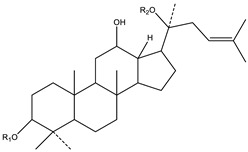	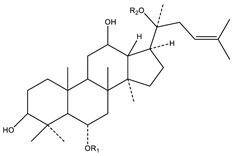
**A. Protopanaxadiol**	**B. Protopanaxatriol**
*R_2_ = Ra_1_, Ra_2_, Ra_3_, Rb_1_, Rb_2_, Rb_3_, Rc, Rd, Rg_2_, Rg_3_, Rs_1_, Rs_2_, etc.*	*R_2_ = Re, Rf, Rg_1_, Rg_2_, Rh_1_, etc.*

**Table 3 biology-09-00252-t003:** Classification of phytoconstituents of *Gymnema sylvestre*.

Phytoconstituents	Classification
Triterpene saponins	Gymnemic acids-acylated (tiglolyl, methylbutyroyl) derivatives of deacylgymnemic acid (DAGA) which is a 3-O-β-glucouronide of gymnemagenin (3β, 16β, 21β, 22α, 23, 28-hexahydroxy-olean-12-ene)
Oleanane saponins	Gymnemic acids and gymnemasaponins
Dammarene saponins	Gymnemosides A, B, C, D, E, and F
Gurmarin	A novel 35-amino-acid peptide with a 4209 molecular weight
Triterpenoidsaponins	
Gymnemasins A	3-O [β-d-glucopyranosyl (1-3)-β-d-glucopyranosyl]-22-O-tiglyol gymnemanol
Gymnemasins B	3-O-[β-d-glucopyranosyl-(1-3)-β-d-glucuro-nopyranosyl]-gymnemanol
Gymnemasins C	glucuronopyranosyl-22-O-tigloyl-gymnemanol
Gymnemasins D	3-O-β-d-glucopyranosyl-gymnemanol
Gymnemanol	3,β-16,β-22, α-23-28-pentahydroxyolean-12-ene
Gymmestrogenin	Pentahydroxytriterpene
Flavonol glycoside	Kaempferol 3-O-β-d-glucopyranosyl-(1-4)-α-l-rhamnopyranosyl-(1-6)- β-d-galactopyranoside
Sterols	Stigmasterol

**Table 4 biology-09-00252-t004:** Some major isolated phytoconstituents on *P. vulgaris*.

Group	Phytochemical Compound
Phenolic acids	Hydroxybezoic acid and derivatives flavonoids, anthocyanins, flavonols, flavanols, isoflavones, flavanones, proanthocyanidins, and tannins
	Hydroxycinnamic acid and derivatives
Flavonoids	Orientin, isoerientin, rutin, myricetin, luteolin, quercetin, kaempferol, myricetin-3-rhamnoside, hyperoside, isorhamnetin-3-glucoside, isoquercitirn
Proteins	Vicilin, phytohenmagglutinin, *alpha*-amylase inhibitor (α-AI1, α-AI2, and α-AIL)

**Table 5 biology-09-00252-t005:** Some major isolated phytoconstituents on *Euphorbia hirta* L.

Group	Compound Name	Structure of Compound
Flavonols	Quercetin	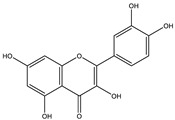
	Kaempferol	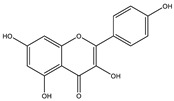
Leucoanthocyanidins	Leucocyanidin	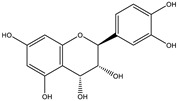
Flavonoid glycosides	Rutin	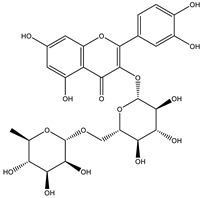
	Quercitrin	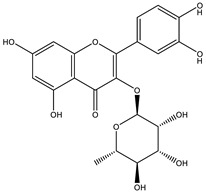
	Myricitrin	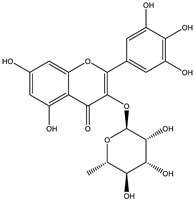
	Afzelin	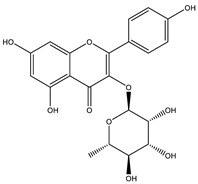
	Luteolin-7-O-glucoside	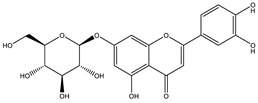
	Isoquercitrin	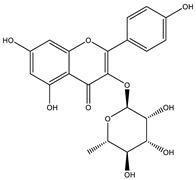
Acids	Syringic acid	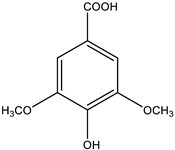
	Ellagic acid	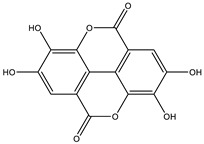
	Gallic acid	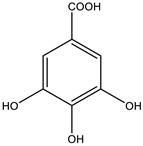
	Shikimic acid	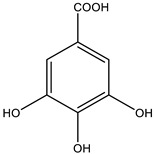
Terpenoids	2β,16α-dihydroxy-ent-kaurane	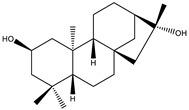
Sterols	β-sitosterol	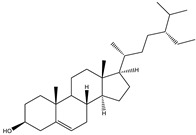
	Stigmasterol	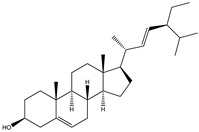

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
