# Peer review of "Bioactive Compounds in Anti-Diabetic Plants: From Herbal Medicine to Modern Drug Discovery"

_biology, 2020, doi:10.3390/biology9090252_

Round 1
Reviewer 1 Report
This paper comprehensively reviewed the active compounds for the treatment of diabetes. Overall, it is a well-prepared article. After some revisions, it may be suitable for publication.
(1) More information should be discussed regarding Gestational Diabetes Mellitus in the Introduction part. It is really a hot-spot issue with less reported articles when compared with investigations of Type I and II diabetes.
(2) Please clarify the specific dose of anti-diabetic bioactive compounds in animals or humans listed in Table 1,2,3,4,5.
(3) The active parts of medicinal plants should be described in Tables
(4) In the part of Conclusion, please discuss the co-regulated targets (if any) among various natural plants
(5) Moderate errors in English writing should be corrected. For example, the sentence in line 16-18 is lack of a verb "Hyperglycemia caused by a deficiency of insulin production by pancreatic (Type 1 diabetes mellitus) or insufficiency of insulin production in the face of insulin resistance (Type 2 diabetes mellitus)."
Author Response
On behalf of the authors, we would like to thank reviewers for your comments which are trully helpful for us to improve the manuscript. Please find our response below
REVIEWER 1
This paper comprehensively reviewed the active compounds for the treatment of diabetes. Overall, it is a well-prepared article. After some revisions, it may be suitable for publication.
(1) More information should be discussed regarding Gestational Diabetes Mellitus in the Introduction part. It is really a hot-spot issue with less reported articles when compared with investigations of Type I and II diabetes.
à We had given more information about GDM
(2) Please clarify the specific dose of anti-diabetic bioactive compounds in animals or humans listed in Table 1,2,3,4,5.
à We had given more details about experiments of investigations above/below tables to clarify information about antidiabetic activity of extracts/bioactive compounds
(3) The active parts of medicinal plants should be described in Tables
à Provided
(4) In the part of Conclusion, please discuss the co-regulated targets (if any) among various natural plants
à Provided
(5) Moderate errors in English writing should be corrected. For example, the sentence in line 16-18 is lack of a verb "Hyperglycemia caused by a deficiency of insulin production by pancreatic (Type 1 diabetes mellitus) or insufficiency of insulin production in the face of insulin resistance (Type 2 diabetes mellitus)."
à Revised
Reviewer 2 Report
Comments to authors and editors:
- The paper is quite long winded and could be extensively reduced in word count.
- It covers general understanding of the problem, some emotional language and also critical reviews of pharmaceuticals and then into individualised chemical components of the herbs.
- What is the single focus?
- If the authors complete a Protocol for Systematic Reviews, with the actual question asked, it may help consolidate the purpose of the paper
- It also does not include all the herbal medicines used for blood glucose control (where the systematic review protocol would help you) so there has to a qualification as to why these herbs are summarised and others are not.
- Maybe the link to the herbal images rather than putting them into the manuscript would help consolidate the information
- For the figures of mechanisms of action, keep these consistent in appearance. But it is a good addition.
- Overall, this is a good draft - but it really needs to be more concise, answering a question that is relevant to the particular audience you are aiming for.
Author Response
On behalf of the authors, we would like to thank reviewers for your comments which are trully helpful for us to improve the manuscript. Please find our response below
Comments to authors and editors:
- The paper is quite long winded and could be extensively reduced in word count.
- We have try to keep it as short as possible
- It covers general understanding of the problem, some emotional language and also critical reviews of pharmaceuticals and then into individualized chemical components of the herbs.
- What is the single focus?
- Our focus is the herb metabolites which lead to its antidiabetes property
- If the authors complete a Protocol for Systematic Reviews, with the actual question asked, it may help consolidate the purpose of the paper
- Our work will be helpful for researchers who work on drug and functional food development from herbal medicine. It takes time to find complete information of a particular herb from reliable resources, especailly herbs’ metabolites.
- It also does not include all the herbal medicines used for blood glucose control (where the systematic review protocol would help you) so there has to a qualification as to why these herbs are summarised and others are not.
- We have try to cover as many as possible and as you can see the paper already quite long. In the future, there might be more new herbs which have similar properties being discovered and new updated review might be needed.
- Maybe the link to the herbal images rather than putting them into the manuscript would help consolidate the information
à We inserted the morphological images to make it convinient for redears.
- For the figures of mechanisms of action, keep these consistent in appearance. But it is a good addition.
- Thanks for your advice
- Overall, this is a good draft - but it really needs to be more concise, answering a question that is relevant to the particular audience you are aiming for.
- We aims to researchers who work on drug and functional food development from herbal medicine.
Round 2
Reviewer 1 Report
Well done. The authors have appropriately revised the manuscript. I have no further comments on it.
Author Response
Thank you for your support!
Reviewer 2 Report
There is still considerable work in correcting grammar, especially in the abstract, introduction and discussion. Many statements are not referred when there should be. It is really important to have another person rad every sentence to check these and also to determine if the sentence was necessary. Your publications are a reflection of you and of the journal and of ourselves, as a reviewer. I suggest that the manuscript is reviewed by a third person, with English grammar as a focus. The content is not in question.
Author Response
Thank you for your comments on our revised manuscript. We have gone through an intensive English proofreading process to make sure there are no typos and grammar errors.
We hope the new version is now suitable for publication.
This manuscript is a resubmission of an earlier submission. The following is a list of the peer review reports and author responses from that submission.